# Adherence to Child Feeding Practices and Child Growth: A Retrospective Cohort Analysis in Cambodia

**DOI:** 10.3390/nu13010137

**Published:** 2020-12-31

**Authors:** Somphos Vicheth Som, Marinka Van Der Hoeven, Arnaud Laillou, Etienne Poirot, Theary Chan, Katja Polman, Maiza Campos Ponce, Frank T. Wieringa

**Affiliations:** 1Section of Infectious Diseases, Department of Health Sciences, Vrije Universiteit Amsterdam, de Boelelaan 1091, 1081 HV Amsterdam, The Netherlands; m.vander.hoeven@vu.nl (M.V.D.H.); c.b.polman@vu.nl (K.P.); m.camposponce@vu.nl (M.C.P.); 2Department of Child Survival and Development, United Nations Children’s Fund Cambodia, Exchange Square Building, Phnom Penh 12101, Cambodia; alaillou@unicef.org (A.L.); epoirot@unicef.org (E.P.); 3Reproductive and Child Health Alliance, Phnom Penh 12100, Cambodia; ctheary@racha.org.kh; 4Department of Biomedical Sciences, Institute of Tropical Medicine, Nationalestraat 155, 2000 Antwerp, Belgium; 5UMR Qualisud, French National Research Institute for Sustainable Development (IRD), IRD/CIRAD/Université de Montpellier/SupAgro, 911 avenue d’Agropolis, 34394 CEDEX 5 Montpellier, France; franck.wieringa@ird.fr

**Keywords:** early child growth, feeding practices, stunting, wasting, concurrent wasting and stunting, longitudinal study, Cambodia

## Abstract

Appropriate feeding in the first 1000 days of a child’s life is critical for their health and growth. We determined associations between adherence to age-appropriate feeding practices and child growth in Cambodia. Children (n = 1079) were included in the first follow-up (FU) data analyses and followed for 30 months (six FUs). Data were analyzed by generalized linear mixed-effect models. Children who adhered to feeding practices on at least three FUs, with an adequate minimal dietary diversity (MDD), a minimal acceptable diet (MAD), and age-appropriate daily feeding (ADF) were less stunted (14.8%, 12.3%, and 6.4%, respectively) than children who never adhered to these indicators (25.2%, 30.1%, and 24.8%, respectively). A higher adherence to MDD and ADF was associated with a higher height-for-age Z-score (HAZ) (β: 0.13, 95% CI: 0.01–0.25 and β: 0.36, 95% CI: 0.22–0.50), while a higher adherence to the MDD and MAD was associated with a higher weight-for-height Z-score (WHZ) (β: 0.19, 95% CI: 0.08–0.30; and β: 0.16, 95% CI: 0.05–0.27). A higher adherence to a minimum meal frequency (MMF) was associated with a lower HAZ (β: −0.99, 95% CI: −1.28–−0.70). Our findings showed that to reduce wasting and stunting in Cambodia, interventions should focus on improving both the quality and quantity of food intake of children under two while targeting the whole complementary feeding period.

## 1. Introduction

The first 1000 days of a child’s life starting from conception until the age of two years is a window of opportunity. It can protect children from the harmful and often irreversible impact of malnutrition on their growth and development [1,2]. Poor child growth is defined by anthropometric indicators such as wasting (weight-for-height Z score <−2 standard deviation (SD)) and stunting (height-for-age Z score <−2 SD) in reference to international standards [3]. Each year, for children less than five years of age, around 800,000 deaths are attributed to wasting and more than one million deaths are attributed to stunting, mostly observed in low- and middle-income countries [4].

These two forms of poor child growth are often identified separately [5], even though it has been recognized that children can be both wasted and stunted at the same time [5,6,7], i.e., concurrent wasting and stunting (WaSt) [5]. The latter condition has received much less attention even though it is more severe and may require special treatment or special attention because of a higher risk of mortality [8]. This division into two forms of malnutrition is also seen in programmatic approaches, with different programs to prevent and treat stunting and wasting [9].

The causes of failure in child growth are complex but, in most cases, have been associated with inadequate feeding practices, infectious diseases, social-economic status, caretaker’s education, and/or sanitation and hygienic practices [10]. The World Health Organization (WHO) has recommended to use infant young child feeding (IYCF) indicators to assess child feeding practices such as exclusive breastfeeding, minimum dietary diversity (MDD), minimum meal frequency (MMF), and minimum acceptable diet (MAD) of a child in the previous 24 h [11]. An MDD is defined as the number of food groups consumed and reflects diet quality [11], an MMF is defined as the number of meals consumed and reflects frequency, and an MAD is a combination of both indicators and reflects both the quality and frequency of food consumed [11]. The Cambodian National Nutrition Program has also recommended the age-appropriate daily feeding (ADF) indicator to assess both the frequency and quantity of food intake among children in the country depending on age [12].

Many earlier studies on IYCF indicators and child growth have shown inconsistent results [13,14,15]. One of the concerns with data on feeding practices is that these often come from cross-sectional studies, with the underlying assumption that caretakers who fed their children appropriately at the time of the study have always been feeding their children appropriately. However, feeding practices may vary during the first two years of life. Therefore, it has been suggested to include information on feeding practices at different points in time in order to better understand associations between feeding practices and child growth [15,16].

Cambodia is one of the low–middle income countries where poor nutrition status remains an important public health problem, with 32.4% of children under five years of age being stunted and 9.6% wasted [17]. In a previous study, we showed that food intake among children aged less than 24 months was insufficient in terms of quality and quantity. More than 70% of the children did not meet MAD requirements [18]. Another study in Cambodia found that ADF showed a positive change in the weight-for-height Z-score (WHZ), but the MDD, MMF, and MAD showed no associations with the change in the WHZ among children aged 0–24 months [19].

To better understand the role of IYCF in growth retardation in Cambodian children, we aimed to determine the association between the adherence to age-appropriate feeding practices (MDD, MMF, MAD, and ADF) and child growth.

## 2. Materials and Methods

### 2.1. Study Design and Sampling

For our study, data from the cohort Cambodia Health and Nutrition Monitoring Study known as “MyHealth” were used. The MyHealth study started in March 2016, and infants and children were recruited into the study until September 2018. Infants and children were seen at regular follow-ups (FUs), with the 1st FU to the 5th FU being conducted every 4 months, while the 6th FU was conducted 14 months after the 5th FU (Table 1). The study was conducted in 6 districts of the 3 provinces in Cambodia, an urban poor area (Phnom Penh), and in rural areas to the northeast (Kratie and Ratanakiri). At the 1st FU, 7738 children aged less than 24 months were randomly selected from the population list provided by Village Health Support Groups (VHSGs). All information on each child’s demography, wealth index, feeding practices, hygiene practices, and anthropometric measurements was collected using electronic tablets. This information was collected by experienced and trained teams.

For our study, only children aged 0–12 months at the 1st FU were selected to investigate their feeding practices for the first year of life. Out of these 2418 children, only those with complete data on feeding practices and anthropometric measurements were included for analysis. Hence, the sample size for the current data analysis was 1079 children in the 1st FU, with a total number of 5984 follow up visits (Figure 1).

### 2.2. Outcome Measures

#### 2.2.1. Child’s Demography

Each child’s age was collected for 30 months from the 1st FU to the 6th FU. The child’s age in months was calculated by subtracting the date of birth of the child from the date of the visit. The child’s age was calculated in months for the continuous variable, or it was categorized into 6 age groups: <6 months, 6–8 months, 9–11 months, 12–23 months, 24–35 months, and 36 months or older. No siblings were included in the current study. Each caretaker’s education was categorized as follows: no education, primary education, secondary education, and high-school or above. Gender was considered as a binary variable (female or male).

#### 2.2.2. Wealth Index

A wealth index indicator was used to reflect household’s socio-economic status (SES) for each child. This indicator was generated using a principal component analysis (PCA), as defined by Filmer and Pritchett [20]. It combined information on a set of material used for housing construction, type of fuel used for cooking, and ownership of household assets such as cars, televisions, generators, radios, computers, bicycles, motorcycles, boats (with and without a motor), phones, and refrigerators. The first principal component was then used to measure the wealth index to calculate the wealth index score [20].

#### 2.2.3. Hygiene Practices

Three main indicators of hygiene practices at the household level were measured through the National-Child Sensitive Composite Score (CSES) defined according to the socio-economic survey and the research brief developed jointly by the Water and Sanitation Program (WSP) and UNICEF [21]. The first one was the main source of water, which was coded as “improved” if it included piped water in dwelling or on premises, a public tap/stand pipe, a tube well or borehole, a protected dug well, and rainwater collection. It was coded as “unimproved” if it included an unprotected dug well, pond, river or stream, vendor-provided water/tanker truck provision, of water or bottled water. The second indicator was safe drinking water, which was classified as “yes” if the water was treated and “no” if the water was not treated. The third one was the type of latrine, which was coded as “improved” if the household had either a pour-flush/flush toilet connected to sewerage, a pour-flush/flush toilet connected to a septic tank, and a pit latrine with slab. Pit latrines without slabs/open pits, a latrine overhanging field/water, public toilet, open land were coded as “not improved.” These methods were used by the National Institute of Statistics of the Ministry of Planning in Cambodia annually to collect information about living condition of the Cambodia population.

#### 2.2.4. Feeding Practices

Children were followed from the 1st FU to the 4th FU within a period of 12 months to assess all age-appropriate feeding practices. The 24-h recall information on child feeding practices was created according to the WHO IYCF indicators for age-appropriate feeding practices [11]. This study followed the breast feeding definition of the WHO: an infant aged 0–59 months who only received breast milk with the exception of vitamins, minerals, medicine, and oral rehydration solution [11]. For exclusive breastfeeding, caretakers were asked at what age did their children started to receive liquids (other than breast milk) or other semi-liquids foods. The MDD indicator included children 6–23 months of age who consumed at least 5 of the following 8 food groups during the previous 24 h: (1) grains, roots, and tubers; (2) legumes and nuts; (3) vitamin A fruits and vegetables; (4) other fruits and vegetables; (5) flesh foods (meats, fish, poultry, and liver/organ meats); (6) eggs; (7) dairy products (milk or yogurt); and (8) breastfeeding. The MMF indicator was fulfilled when either breastfed or non-breastfed children 6–23 months of age consumed solid food, semi-solid food, or milk feeds at least the minimum recommended number of times during the previous 24 h. An MMF was defined as 2 mealtimes for breastfed children at 6–8 months of age, 3 mealtimes for breastfed children at 9–23 months of age, and 4 mealtimes for non-breastfed children 6–23 months of age. The MAD indicator was calculated based on the following criteria: (1) breastfed children 6–23 months of age who had at least an MDD and an MMF and (2) non-breastfed children 6–23 months of age who received at least 2 milk feedings and had at least an MDD and an MMF.

The ADF indicator was created according to the Cambodian National Nutrition Program recommendation based on the children’s age groups presented in Table 2 [12]. This indicator was assessed based on 24-h recall information of feeding practices to measure the frequency and quantity of breastfeeding and complementary feeding. The quantity of complementary feeding was measured using tablespoons or traditional bowls (Chan Chang Koeh). It was assumed that 1 tablespoon equals 10 mL and 1 Chan Chang Koeh equals 250 mL. It was also assumed that an energy density of 1 Chan Chang Koeh equals 150 Kcal and 1 tablespoon equals 6 Kcal.

To assess the adherence to age-appropriate feeding practices, the adherence to each of the four age-appropriate feeding indicators was determined for each of the four FUs in a 12 months period. Adherence was scored as follows: fulfillment of indicator per FU = 1, and no fulfilment of indicator per FU = 0. The sum of the four scores per indicator could therefore range from 0 to 4. The sum of each indicator was grouped into three categories: never, 1–2 times, and 3 times to always. The sum of the MDD, MMF, and MAD indicators for children 0–5 months of age was coded as follows: exclusive breastfeeding = 1, and not exclusively breastfed = 0.

#### 2.2.5. Anthropometric Measurements

Anthropometric measurements were collected for 30 months from the 1st FU to the 6th FU. The measurements were performed following standard procedures [22] by trained investigators. All anthropometric measurements, including weight and height, were measured twice per child to ensure measurement accuracy. Weight was measured using calibrated digital scales (SECA 874, Hamburg, Germany) with 100 g of precision. Caretakers of children were asked to remove their child’s thick clothes or remove their child’s diaper before the measurement was performed. Recumbent length or standing height was measured to the nearest 1 mm using UNICEF boards.

All anthropometric variables were calculated according to the WHO Child Growth Standards of a height-for-age Z-score (HAZ) and a WHZ [23]. Wasting was defined as a WHZ <−2 SD, stunting was defined as an HAZ <−2 SD, and children that were both wasted and stunted were categorized as “WaSt.” To ensure the accuracy of the anthropometric variables, extreme values were checked. If an HAZ <−6 or >5 or if a WHZ <−5 or >5, they were excluded from the analyses.

### 2.3. Statistical Analysis

All data analysis was done using STATA software version 14 (Stata Corp LLC., College Station, TX, USA). Descriptive analysis was performed to describe variables with a normal distribution using the mean, standard deviation, and range. Categorical variables were expressed as frequencies and percentages. A chi-square test were used to analyze the association between growth status and age, between age-appropriate feeding practices and age, and between growth status and adherence to age-appropriate feeding practices.

To determine the relationship of adherence to age-appropriate feeding indicators and other potential determinants with child’s growth, generalized linear mixed-effects models were done. Linear mixed-effects were used for continuous outcomes (HAZ and WHZ), and mixed-effects logistic regression was done for binary outcome (WaSt). Selected indicators for fixed effects included: regions, age, gender, age-appropriate feeding practices, caretaker’s education level, wealth index, child’s nutritional status, household main drinking water source, treatment of drinking water, and ownership of latrine. The random effects were used the child’s ID with random slope of child’s age and nested by regions. Since there is a non-linear relationship between age and child growth, age square was added into the model. Forward selection was performed, and variables with a *p*-value < 0.1 in a bivariate analysis were included into the multivariable model. The fitted models were compared by using the Akaike information criterion (AIC) or Bayesian information criterion (BIC) estimates of model performance [24]. An identity correlation matrix was chosen for the correlation among repeated child’s nutritional status measurements [25].

Sensitivity analyses were performed to check the validity and consistency of the models. The selection criteria for sensitivity analyses were similar as our main analyses. The difference between them is that we included children for whom data collection regarding the feeding practices was not fully complete (one or more follow-ups were missing). Generalized linear mixed-effects models were done to determine the association between adherence to age-appropriate feeding practices and child growth. Fixed effects included regions, age, gender, age-appropriate feeding practices, caretaker’s education level, wealth index, child’s nutritional status, household main drinking water source, treatment of drinking water, and ownership of latrine. The child’s ID with random slope of child’s age and nested by regions were selected as random effects. Age square was added into the models. Variables with *p*-value < 0.1 were selected in the full model. The fitted models were compared by using the AIC or BIC estimates of model performance. An identity correlation matrix was chosen for the correlation among repeated child’s nutritional status measurements.

### 2.4. Ethics Statement

The MyHealth study received ethical approval by the National Ethics Committee for Health Research of the Ministry of Health of Cambodia; the file code number is 117/NECHR on 28 May 2016. The study objectives were explained to parents or caretakers of the children, and written consent was requested from them before any data were collected. Participation in the study was voluntary, and parents or caretakers could withdraw their child from the study at any time without any consequences.

## 3. Results

### 3.1. Background Characteristics

At baseline, the number of children recruited from each of the three regions were similar. The mean age of children at baseline was 6.6 (±3.5 SD) months (Table 3). The percentages of girls and boys were almost equal (50.7% of girls and 49.3% of boys). Most caretakers in our study had completed primary education (45.7%), and only 13.1% completed high school or higher education (Table 1). Most households had an improved main water source (73.1%) and treated their drinking water (77.3%); however, only half of these households had an improved toilet.

### 3.2. Childgrowth Status

The percent of stunting almost doubled from 0–5 months of age to 9–11 months of age (12.8%–23.2%), and it kept increasing with a peak of 38.9% among children aged 24–35 months (Figure 2). The percent of wasting was consistent (~10%) across all age groups. The percentage of WaSt increased from 0.7% among children aged less than 6 months to 7.0% among children aged 24–35 months.

### 3.3. Feeding Practices

The percentage of age-appropriate feeding practices varied across age groups. The percentage of children that had received age-appropriate MMF and ADF values was highest among children aged six-to-eight months (90.8% and 64.6%, respectively) and lowest among children aged 12–23 months (70.8% and 16.1%, respectively) (Figure 3). In contrast, the percentages of children that had received an age-appropriate MDD and MAD was lowest in the youngest children aged six-to-eight months (20.3% and 19.3%, respectively) and highest in older children aged 12–23 months (40.4% and 32.8%, respectively) (Figure 3).

Figure 4 shows the percentage of adherence to age-appropriate feeding practices. The percentages of children who adhered to at least three times the MDD, MAD, and ADF were 18.3%, 13.5%, and 7.0%, respectively.

Adherence to the IYFC indicators was associated with a better nutritional status (Figure 5). Children who adhered at least three times to the MDD, MAD, and ADF were less likely to be stunted (14.8%, 12.3%, and 6.4%, respectively) compared to those children who never adhered to these feeding practices (25.2%, 30.1%, and 24.8%, respectively). However, children who adhered at least three times to the MMF showed a higher percent of stunting, wasting, and WaSt (65.7%, 70.5%, and 68.4%, respectively) compared to children who never adhered to the MMF (1.4%, 0.8%, and 0.4%, respectively) (Figure 5).

### 3.4. Relationship of Adherence to Age-Appropriate Feeding Practices and Other Potential Determinants with Child Growth

Linear mixed effect regression modelling showed that children with a higher adherence to the MDD had a higher HAZ (adhered one-to-two times to the MDD = β: 0.12, 95% CI: 0.03–0.22; and adhered to three times or more to the MDD = β: 0.13, 95% CI: 0.01–0.25) compared to children who never adhered to the MDD (Table 4). Additionally, children who adhered at least three times to ADF had a significantly higher HAZ (β: 0.36, 95% CI: 0.22–0.50) than children who never adhered to ADF (Table 4). Children who were adherent to the MMF more times had a lower HAZ (adhered one-to-two times to the MMF = = β: −0.95, 95% CI: −1.24–−0.66; and adhered three times or more to the MMF = β: −0.99, 95% CI: −1.28–−0.70) compared to children who never adhered to the MMF. Meanwhile, adherence to the MAD did not show any significant association with the HAZ (Table 4). Other factors such as child’s sex, child’s age, caretaker’s education, wealth index, treatment of drinking water, and having improved toilet were strong predictors of a child’s HAZ (Table 4). The results of the sensitivity analysis were similar to our findings; however, only children who adhered to one-to-two times to the MDD had a significantly higher HAZ (shown in Appendix A).

In Table 5, the results of the mixed effect linear regression model on adherence to age-appropriate feeding practices and the child’s WHZ are presented. Only children who adhered at least three times to the MDD had a higher WHZ (β: 0.19, 95% CI: 0.08–0.30). Children with a higher adherence to the MAD had a higher WHZ (β: 0.16, 95% CI: 0.05–0.27) than those children who never adhered to the MAD. Adherence to the MMF and ADF was not associated with the WHZ. Wealth index and child’s age were the strongest predictors of a child’s WHZ, while caretaker’s education showed the least effect on a child’s WHZ in the full adjusted model (Table 5). The sensitive analyses showed similar findings (shown in Appendix A).

In Table 6, the results of the mixed effect logistic regression model on adherence to age-appropriate feeding practices and the child’s WaSt are presented. Adherence to the MDD, MMF, MAD, and ADF had no significant association with WaSt in the full adjusted model (Table 6).The sensitivity analyses also found that none of the age-appropriate feeding practices had association with WaSt (shown in Appendix A).The model also showed that wealth index was the strongest predictor for WaSt (AOR: 0.58, 95% CI: 0.43–0.77), and caretaker’s education was the second strongest predictor for WaSt (AOR: 0.88, 95% CI: 0.81–0.97). Children aged 36 months or older were less likely to be WaSt than children aged from zero to less than six months (Table 6).

## 4. Discussion

To our knowledge, this has been one of the first longitudinal studies on the association between age-appropriate feeding practices with child growth, and it has been the first study to use the sum scores per each indicator of age-appropriate feeding practices to capture a wide range of information on feeding practices during the first 1000 days life in Cambodia. We found that children with a higher adherence to adequate quality and quantity (MDD and ADF) of food intake had a higher HAZ. The percentages of stunting, wasting, and WaSt were lower among children with higher adherences to adequate quality and quantity of food intake (MDD, MAD, and ADF) compared to those children with no adherence. Similar to our findings, the study of association of IYCF with linear growth counted in 11 countries also found that children with an adequate MDD were less likely to be stunted in Indonesia and that the MDD was positively associated with the HAZ in India, Bangladesh, and Zambia [26]. Our study also found that the percentage of stunting, wasting, and WaSt was lower among children who never adhered to an adequate meal frequency (MMF) compared to those children who did adhere. Children with a higher adherence to an adequate meal frequency (MMF) had a lower HAZ. Our study suggests that to sustain linear growth in Cambodian children, the quality and quantity of the diet are more important than meal frequency. Interestingly, from our data, it appears that for sustaining ponderal growth, both quality and meal frequency are important, as only an MDD and an MAD were associated with a higher WHZ. Though the findings showed an association, there was no causality between adherences to age-appropriate feeding practices and child growth.

Indeed, as we found, we were not the first to report that the MMF is not a good indicator for linear growth. A study on pooled DHS data from 14 countries and a study from Zambia reported found that only children with an adequate quality of food intake (MDD) had a lower probability of stunting, while an adequate frequency of food intake (MMF) showed no effect [27,28]. These results suggest that providing a sufficient quality and amount of food is critical for child growth. Studies on food intake among children in Cambodia have reported that children have received insufficient quantity and quality of food [29,30]; for example, children have received rice porridge mostly containing water and rice without other ingredients [31]. This suggests that interventions and programs that are targeted to improve child growth should place more emphasis on food quality and quantity rather than on meal frequency.

Recent studies have suggested an interaction between poor child growth and environmental enteric dysfunction (EED) [32,33], which refers to malfunctioning of the small intestine that results in inflammation and reduced surface area of intestinal villi, thus causing nutrient malabsorption [34]. Children in Cambodia are mostly living in a poor environment. As our data showed, almost 50% were reported as not having a toilet, and 30% of them were reported as not treating their drinking water. These children may often be exposed to repeated infections that result in EED [33], which could be one explanation for why the MMF showed a reversed direction association with the HAZ.

Our study showed that the percent of sub-optimal linear and ponderal growth differed across age groups. The percentage of stunting and WaSt was higher among older children (24–35 months of age). These findings were similar to the findings of the Cambodian DHS in 2014 [17]. However, they showed that the percentage of stunting started to decline only after 48 months of age, while in our study, the decline in stunting percentage had already started at 35 months of age. This apparent ‘catch-up’ in linear growth was also noted before in a large cohort study on children in the Gambia [35]. As the extent of catch-up linear growth was observed to decline after 35 months of age, it is worth raising the question of whether the ‘1000-days window’ concept is too narrow, as well as whether interventions outside of this time-frame might be needed to support the optimal growth of children.

Indeed, data on both linear and ponderal growth support this. A meta-analysis of WaSt among children aged 6–59 months from 84 countries showed that the highest prevalence of WaSt was found in children aged 24–35 months, similar to our findings [36]. This suggests that more effort is needed to combat poor growth among children under five years of age. Recent findings from the MyHealth study showed that maternal nutritional status is an important factor in early child growth, and interventions aimed to combat poor growth should not be restricted to children less than 24 months of age [37]. At the same time, to combat both wasting and stunting in early childhood, continuous efforts to improve both the dietary quality and quantity of food intake are needed over the whole complementary feeding period, and short interventions might not be sufficient.

Interestingly, there were differences in the reasons for inadequate age-appropriate feeding practices across the age groups. Infants received diets that lacked quality and frequency (MDD and MAD), while older children (12–23 months of age) had diets that lacked in quantity and frequency (ADF and MMF). Similar to our findings, a study on the relationship of child feeding and growth in 14 countries found that food quality and frequency (MDD and MAD) were often better among older children [27].

The full adjusted models also showed that the child’s sex, child’s age, wealth index, caretaker’s education, and hygiene practices were significantly associated with the HAZ. Similar to our results, a study in Cambodia found that access to improved water source and sanitation were significantly associated with a child’s HAZ [38]. A child’s age caretaker’s education, and wealth index were associated with their WHZ and WaSt. However, hygiene practices showed no effect on the WHZ or WaSt. This finding suggests that there is a need for integrated strategies from different sectors to combat stunting, wasting, and WaSt in Cambodia.

One limitation of our study was the data selection, since we only selected children who fully attended the first four FUs of the data collection. Hence, our results might have underestimated or were unable to capture the effect of feeding practices among those caretakers who were lost to follow-up. Those children are likely to be the ones with a poor growth status and/or a poor adherence to age-appropriate feeding practices. Another limitation is the lack of information on fetal growth, which was outside of the scope of the overarching study. As known, poor fetal growth can have long-lasting effects on post-natal growth despite providing optimal nutrition after delivery [39,40]. Additionally, we did not include EED in our study because it is one of the potential determinants for child growth [33]. Lastly, child growth causes are complex, and some of those factors that could have been potential confounders were beyond the scope of our study.

## 5. Conclusions

Our study showed that providing adequate feeding for children younger than two years old and for the whole complementary feeding period is needed to achieve optimal growth. Causes of inadequate feeding practices (insufficient quality, quantity, and frequency), including breastfeeding, differed among age groups and for linear and ponderal growth. Therefore, we suggest that any intervention or program aiming to reduce the prevalence of poor child growth in Cambodia should be carried out using different approaches that should place greater emphasis on quality and quantity for both breastfeeding and complementary feeding based on a child’s age. Furthermore, collaboration with other sectors to improve caretaker’s education, household income, and hygiene practices is needed to make a sustainable impact on child growth in Cambodia.

## Figures and Tables

**Figure 1 nutrients-13-00137-f001:**
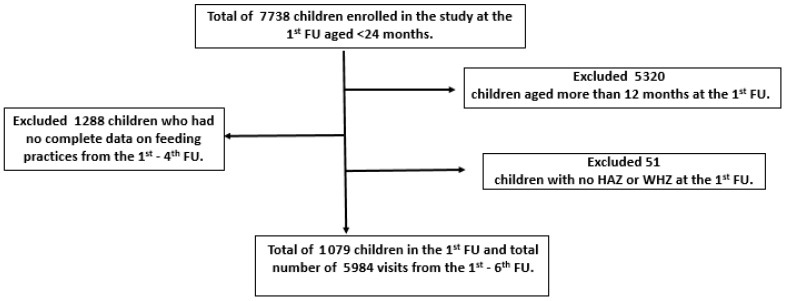
Flow diagram of the current study. FU = follow-up; HAZ = height-for-age Z-score; WHZ = weight-for-height Z-score.

**Figure 2 nutrients-13-00137-f002:**
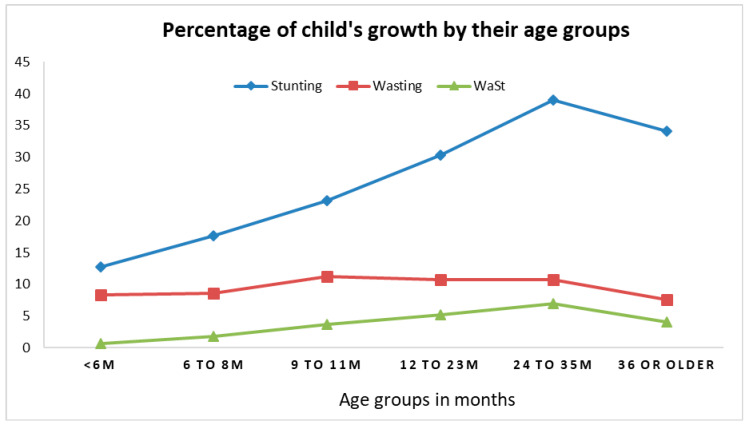
Percentage of child growth by age group from the 1st to 6th follow-up (n = 1079; observations = 5984). Stunting = height-for-age Z score <−2 standard deviation; Wasting = weight-for-height Z score <−2 standard deviation; and WaSt = concurrence wasting and stunting.

**Figure 3 nutrients-13-00137-f003:**
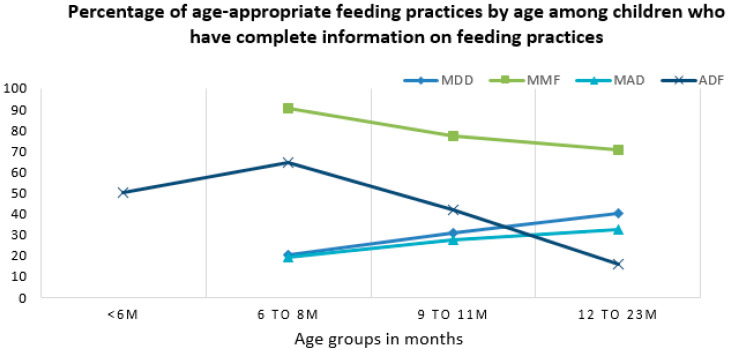
Percentage of age-appropriate feeding practices by child’s age groups. MDD = minimum dietary diversity; MMF = minimum meal frequency; MAD = minimum acceptable diet; and ADF = age-appropriate daily feeding.

**Figure 4 nutrients-13-00137-f004:**
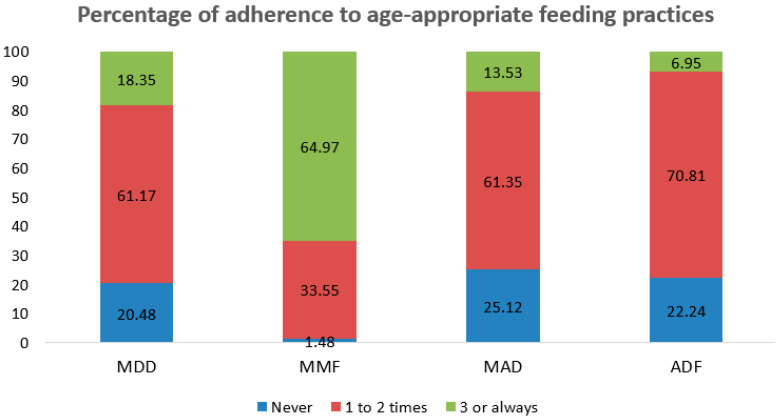
Percentage of adherence to age-appropriate feeding practices in children aged 0–23 months from the 1st to 4th follow-up. MDD = minimum dietary diversity; MMF = minimum meal frequency; MAD = minimum acceptable diet; and ADF = age-appropriate daily feeding.

**Figure 5 nutrients-13-00137-f005:**
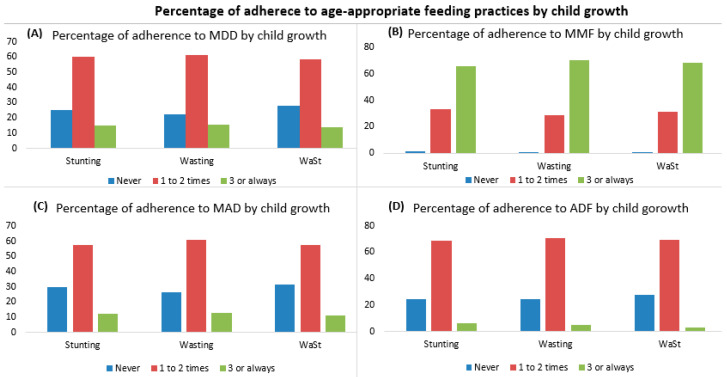
Percentage of adherence to age-appropriate feeding practices among children aged 0–23 months by their growth from the 1st to 6th follow-up. Stunting = height-for-age Z score <−2 standard deviation; Wasting = weight-for-height Z score <−2 standard deviation; and WaSt = concurrence wasting and stunting. MDD = minimum dietary diversity; MMF = minimum meal frequency; MAD = minimum acceptable diet; and ADF = age-appropriate daily feeding. (**A**) Percentage of total number of adherence to MDD by child growth; (**B**) Percentage of total number of adherence to MMF by child growth; (**C**) Percentage of total number of adherence to MAD by child growth; and (**D**) Percentage of total number of adherence to ADF by child growth.

**Table 1 nutrients-13-00137-t001:** Timeline of follow-ups (FUs) for the study.

	2016									2017						2018		
	1st FU			2nd FU			3rd FU			4th FU			5th FU			6th FU		
	Mar	Apr	May	Jun	Jul	Aug	Oct	Nov	Dec	Feb	Mar	Apr	Jun	Jul	Aug	Jun	Aug	Sep
	*PP*	*KR*	*RT*	*PP*	*KR*	*RT*	*PP*	*KR*	*RT*	*PP*	*KR*	*RT*	*PP*	*KR*	*RT*	*RT*	*KR*	*PP*
Anthropometric measurement	x	x	x	x	x	x	x	x	x	x	x	x	x	x	x	x	x	x
Feeding practices	x	x	x	x	x	x	x	x	x	x	x	x						

PP = data collection was conducted in Phnom Penh; KR = data collected was conducted in Kratie; and RT = data collection was conducted in Ratanakiri. FU = follow-up. X = presenting the availability of the data collection for each follow-up.

**Table 2 nutrients-13-00137-t002:** The construction of age-appropriate daily feeding (ADF).

Age Groups (in Months)	Breast Feeding	Complementary Feeding	
		Frequency	Quantity ^1^
0–5	Predominant breastfeeding at least 6 times per day	None	None
6–8	Breastfeeding at least 8 times per day	Maximum of 3 meals per day	2 to 3 tablespoons up to at least half a Chan Chang Koeh for each meal
9–11	Breastfeeding at least 6 times per day	Minimum of 3 meals per day	At least half up to a full Chan Chang Koeh for each meal
12–24	Breastfeeding at least 6 times per day	Minimum of 3 meals per day	At least half up to a full Chan Chang Koeh for each meal

^1^ It was assumed that an an energy density of 1 Chan Chang Koeh equals 150 Kcal and 1 table spoon equals 6 Kcal [12]. The table was adapted from a previous study [19].

**Table 3 nutrients-13-00137-t003:** Background characteristics of participants at 1st follow-up.

	*n*	%
Regions	N = 1079	
Ratanakiri	349	23.3
Kratie	407	37.7
Phnom Penh	323	29.9
Sex	N = 1079	
Girls	547	50.7
Age at 1st follow-up (in months)	N = 1079	
0–5	493	45.7
6–8	280	26.0
9–11	242	22.4
12–23	64	5.9
Caretaker’s education	N = 1067	
No education	200	18.7
Primary	488	45.7
Secondary	239	22.4
High-school or above	140	13.1
Socio-economic status	N = 1064	
Poorest	183	17.2
Poor	237	22.3
Middle	267	25.1
Rich	161	15.1
Richest	216	20.3
Hygiene practices	N = 1025	
Household main drinking water sources (improved)	749	73.1
Drinking water was treated	792	77.3
Improved toilet	537	52.4

N = Total number of participants in the 1st follow-up.

**Table 4 nutrients-13-00137-t004:** The relationship of adherence to age-appropriate feeding practices and other determinants with child height-for-age (HAZ).

HAZ									
	Model 1 ^a^			Model 2 ^b^			Model 3 ^c^		
Full Adjusted Model	Total Number of Adherence to MDD or MMF (n = 1018; Observations = 5647)			Total Number of Adherence to MAD (n = 1018; Observations= 5647)			Total Number of Adherence to ADF (n = 1018; Observations = 5647)		
Fixed Effects	Cof	95 CI%	*p*-Value	Cof	95 CI%	*p*-Value	Cof	95 CI%	*p*-Value
Sex (boy)	−0.10	0.16–−0.03	0.006	−0.09	−0.16–−0.02	0.012	−0.09	−0.16–−0.02	0.008
Age (in month)	−0.09	−0.10–−0.08	<0.001	−0.09	−0.10–−0.09	<0.001	−0.09	−0.10–−0.08	<0.001
Age^2^ (in month)	0.002	0.001–0.002	<0.001	0.002	0.001–0.002	<0.001	0.002	0.001–0.002	<0.001
Total number of adherence to MDD (Never)									
1 or 2 times	0.12	0.03–0.22	0.013						
3 or always	0.13	0.01–0.25	0.032						
Total number of adherence to MMF(Never)		Reference							
1 or 2 times	−0.95	−1.24–−0.66	<0.001						
3 or always	−0.99	−1.28–−0.70	<0.001						
Total number of adherence to MAD (Never)									
1 or 2 times				0.01	−0.08–0.10	0.787			
3 or always				−0.03	−0.15–0.09	0.590			
Total number of adherence to ADF(Never)									
1 or 2 times							0.01	−0.07–0.10	0.758
3 or always							0.36	0.22–0.50	<0.001
Maternal education (years)	0.02	0.01–0.03	<0.001	0.02	0.01–0.03	<0.001	0.02	0.01–0.03	<0.001
Wealth index	0.11	0.08–0.14	<0.001	0.11	0.08–0.14	<0.001	0.12	0.09–0.15	<0.001
Caretaker drinking water was treated	0.13	0.04–0.22	0.007	0.14	0.04–0.23	0.004	0.13	0.04–0.22	0.007
Improved toilet	0.09	0.01–0.17	0.029	0.10	0.02–0.18	0.016	0.10	0.02–0.18	0.015
Random effect									
Variances of random intercept	0.04	0.007–0.19		0.04	0.01–0.21		0.04	0.01–0.22	
Variance of random slope	0.002	0.001– 0.002		0.002	0.001–0.002		0.002	0.001–0.002	
Variance of residuals	0.53	0.51–0.56		0.54	0.52–0.56		0.53	0.51–0.56	
Log-likelihood		−7294.99			−7317.73			−7301.941	
LR test		<0.001			<0.001			<0.001	

^a^ Model 1 was a linear mixed-effects model of the relationship between adherence to MDD or MMF and other determinants with the HAZ. ^b^ Model 2 was a linear mixed-effects model of the relationship between adherence to MAD and other determinants with the HAZ. ^c^ Model 3 was a linear mixed-effects model of the relationship between adherence of ADF and other determinants with the HAZ. Age^2^ = Age square is added to the model since the relationship between age and HAZ is non-linear. *p*-Value is statistically significant at *p* < 0.05. MDD = minimum dietary diversity, MMF = minimum meal frequency; MAD = minimum acceptable diet; and ADF = age-appropriate daily feeding; Cof = coefficient; CI = confidence interval.

**Table 5 nutrients-13-00137-t005:** The relationship of adherence to age-appropriate feeding practices and other determinants with child Weight-for-Height (WHZ).

WHZ									
	Model 1 ^a^			Model 2 ^b^			Model 3 ^c^		
Full Adjusted Model	Total Number of Adherence to MDD or MMF (n = 1058; Observations = 5871)			Total Number of Adherence to MAD (n = 1058; Observations = 5871)			Total Number of Adherence to ADF (n = 1058; Observations = 5871)		
Fixed effects	Cof	95 CI%	*p*-Value	Cof	95 CI%	*p*-Value	Cof	95 CI%	*p*-Value
Age (in month)	−0.05	−0.06–−0.05	<0.001	−0.05	−0.06–−0.04	<0.001	−0.05	−0.06–−0.05	<0.001
Age^2^ (in months)	0.001	0.0009–0.0013	<0.001	0.001	0.0009–0.0013	<0.001	0.001	0.0009–0.0013	<0.001
Total number of adherence to MDD (Never)									
1 or 2 times	0.05	−0.04–0.13	0.318						
3 or always	0.19	0.08–0.30	0.001						
Total number of adherence to MMF(Never)									
1 or 2 times	0.20	−0.07–0.47	0.155						
3 or always	0.07	−0.20–0.34	0.634						
Total number of adherence to MAD (Never)									
1 or 2 times				0.05	−0.03–0.13	0.215			
3 or always				0.16	0.05–0.27	0.005			
Total number of adherence to ADF(Never)									
1 or 2 times							−0.02	−0.10–0.06	0.632
3 or always							0.01	−0.13–0.14	0.928
Maternal education (years)	0.01	−0.00–0.02	0.052	0.01	0.0002–0.02	0.045	0.01	0.001–0.02	0.030
Wealth index	0.07	0.047–0.100	<0.001	0.08	0.05–0.10	<0.001	0.08	0.05–0.10	<0.001
Random effects									
Variances of random intercept	0.01	0.0009–0.04		0.006	0.001–0.04		0.01	0.001–0.04	
Variance of random slope	0.002	0.001–0.002		0.001	0.001–0.002		0.002	0.001–0.002	
Variance of residuals	0.55	0.53–0.58		0.56	0.53–0.58		0.56	0.53–0.58	
Log-likelihood		−7514.27			−7521.16			−7524.93	
LR test		<0.001					<0.001	<0.001	

^a^ Model 1 presents linear mixed-effects model of the relationship between adherence to MDD or MMF and other determinants with WHZ. ^b^ Model 2 presents linear mixed-effects model of the relationship between adherence to MAD and other determinants with WHZ. ^c^ Model 3 presents linear mixed-effects model of the relationship between adherence to ADF and other determinants with WHZ. Age^2^ = Age square is added to the model since the relationship between age and WHZ is non-linear. *p*-Value is statistically significant at *p* < 0.05. MDD = minimum dietary diversity, MMF = minimum meal frequency; MAD = minimum acceptable diet; and ADF = age-appropriate daily feeding; Cof = coefficient; CI = confidence interval.

**Table 6 nutrients-13-00137-t006:** The relationship of adherence to age-appropriate feeding practices and other determinants with child concurrence wasting and stunting (WaSt).

WaSt									
	Model 1 ^a^			Model 2 ^b^			Model 3 ^c^		
Full Adjusted Model	Total Number of Adherence to MDD or MMF (n = 1058; Observations = 5871)			Total Number of Adherence to MAD (n = 1058; Observations = 5871)			Total Number of Adherence to ADF (n = 1058; Observations = 5871)		
Fixed effect	AOR	95 CI%	*p*-Value	AOR	95 CI%	*p*-Value	AOR	95 CI%	*p*-Value
Age (in month)	1.32	1.22–1.44	<0.001	1.32	1.21–1.44	<0.001	1.32	1.21–1.44	<0.001
Age^2^ (in months)	0.99	0.992–0.996	<0.001	0.99	0.992–0.996	<0.001	0.99	0.992–0.996	<0.001
Total number of adherence to MDD (Never)		Reference							
1 or 2 times	0.83	0.38–1.86	0.67						
3 or always	0.61	0.19–1.89	0.39						
Total number of adherence to MMF(Never)									
1 or 2 times	2.98	0.09–101.60	0.55						
3 or always	3.48	0.10–117.13	0.48						
Total number of adherence to MAD (Never)									
1 or 2 times				0.85	0.40–1.80				
3 or always				0.68	0.21–2.20				
Total number of adherence to ADF(Never)									
1 or 2 times							1.04	0.49–2.22	0.92
3 or always							0.26	0.044–1.59	0.15
Maternal education (years)	0.86	0.77–0.95	0.003	0.86	0.77–0.95	0.003	0.85	0.77–0.94	0.002
Wealth index	0.58	0.43–0.77	<0.001	0.57	0.42–0.76	<0.001	0.56	0.42–0.75	<0.001
Random effect									
Variances of random intercept	8.45	5.84–12.23		8.52	5.89–12.32		8.46	5.86–12.23	
Log-likelihood		−774.68			−775.07			−773.80	
LR test			<0.001			<0.001			<0.001

^a^ Model 1 presents mixed-effects logistic regression of the relationship between adherence to MDD or MMF and other determinants with WaSt. ^b^ Model 2 presents mixed-effects logistic regression of the relationship between adherence to MAD and other determinants with WaSt. ^c^ Model 3 presents mixed-effects logistic regression of the relationship between adherence to ADF and other determinants with WaSt. Age^2^ = Age square is added to the model since the relationship between age and WaSt is non-linear. *p*-Value is statistically significant at *p* < 0.05. MDD = minimum dietary diversity, MMF = minimum meal frequency; MAD = minimum acceptable diet; and ADF = age-appropriate daily feeding; AOR = adjusted odds ratio; CI = confidence interval.

## Data Availability

The data presented in this study are openly available at https://doi.org/10.3390/nu13010137.

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
