# Peer review of "Adherence to Child Feeding Practices and Child Growth: A Retrospective Cohort Analysis in Cambodia"

_nutrients, 2020, doi:10.3390/nu13010137_

Round 1

Reviewer 1 Report

Based on a cohort of 1079 children who were followed for 30 months (6 follow-ups), the study aimed to determine associations between adherence to age-appropriate feeding practices and child growth in Cambodia. Main outcomes were stunting and concurrent wasting and stunting. It was concluded that child’s sex, child’s age, wealth index, caretaker’s education, hygiene practices were also associated with HAZ 34 while child’s nutritional status, child’s age wealth index and caretaker’s education were associated with WHZ.

Below are my comments:

  1. Abstract. (1) Please make clear the design of this study. (2) Provide a brief description of statistical model for calculating the beta and its 95%CI.
  2. Materials and Methods

Please make clear “the Cambodia Health and Nutrition 86 Monitoring Study” is a cohort study.

  1. Statistical analysis

(1) The term “longitudinal mixed effect linear regression” is rarely used in literature. You probably mean the generalised linear mixed model. Please be precise about this model is and what the fixed and random effects are.

(2) Figure 2 shows clear nonlinear age effect. How nonlinearity was addressed in your statistical analysis?

(3) Table 1 shows there are missing data in some of baseline characteristics of participants. Please describe your strategy for dealing with missing data.

(4) Age-appropriate feeding practices were measured after measurement of children’s HAZ and WHZ, resulting in the so-called reverse causality issue and making the results of mixed model analysis difficult to interpret.

(5) Please describe your sensitivity analysis here.

  1. Results

(1) Is there any interaction between age and feeding practices as Figure 3 suggests that feeding practices change with age.

(2) This is a cohort study but occurrences of stunting, wasting and WaSt are interpreted as prevalence. I think it should be interpreted as incidence and prevalence is used for cross-sectional studies.

  1. Table 2

(1) Please provide the number of observations used in each models. I guess they are different.

(2) Age is treated as a linear effect but Figure 2 suggests not.

(3) If P=0.000, please report as P<0.001

  1. Table 3

See my comments on Table 2 and they apply here too.

  1. Table 4

See my comments on Table 2 and they apply here too.

  1. Discussion

(1) Please move the results of sensitivity analysis into Results section.

(2) Please mention the unobserved confounding issue and possible reversed causality issue in the limitations.

Author Response

We would like to thank the reviewer for their time on reviewing our manuscript and providing useful comments and suggestions. Please find our responses point-by-point as following:

Response to Reviewer #1

Point 1: Abstract. (1) Please make clear the design of this study. (2) Provide a brief description of statistical model for calculating the beta and its 95%CI.

Response 1: The description of the study design and the statistical models for calculating the beta and its 95% CI were added to the abstract as suggestion in Line 23 and 2-24. Due to word limitation criteria, we made more change in the abstract. 

Point 2: Materials and Methods

Please make clear “the Cambodia Health and Nutrition 86 Monitoring Study” is a cohort study

Response 2: The cohort study was added in Line 100 to clarify the design of “The Cambodia Health and Nutrition Monitoring Study” as suggested by the reviewer.

Point 3: Statistical analysis.

Point 3.1: The term “longitudinal mixed effect linear regression” is rarely used in literature. You probably mean the generalised linear mixed model. Please be precise about this model is and what the fixed and random effects are.

Response 3.1: The term “longitudinal mixed effect linear regression” was changed to “the generalized linear mixed-effect models” in line 226. Both fixed effects and random effects of the generalized linear mixed-effect models were added as suggested in line 227. Both fixed effect variables and random effect are clearly describe in line 229-233.

Point 3.2: Figure 2 shows clear nonlinear age effect. How nonlinearity was addressed in your statistical analysis?

Response 3.2: We thank for your comments on the nonlinear age effect. Since the relationship between child growth and age is nonlinear, we now included a quadratic term for age (age square) in our generalized linear mixed effects models.

Point 3.3: Table 1 shows there are missing data in some of baseline characteristics of participants. Please describe your strategy for dealing with missing data.

Response 3.3: Originally, the data files were arranged separately: (1) child’s background (regions, age, sex, and nutritional status) file, (2) caretaker’s education, (3) socio-economic file and (4) Hygiene practice files. All those files contained unique child’s ID. The files were then merged into a single file using child’s ID to match the data. However, some of the child’s ID were incorrectly entered by our data collectors in one file resulting in missing data. Due to this reason, we may be able to conclude that our missing data was missing at random. In the linear mixed effect and mixed effect logistic regress, these missing values were then drop from the analyses. This is the reason why in our full adjusted models contain less number of children and observations. This resulted in at most 5% of children with missing values in the analyses. Moreover, for the main indicators (adhering to feeding practices) and main outcomes (child nutritional status) we did not have missing values. Background characteristics, caretakers, education, wealth index and WASH are added as others confounders for the association between feeding practices and child growth. 

Point 3.4: Age-appropriate feeding practices were measured after measurement of children’s HAZ and WHZ, resulting in the so-called reverse causality issue and making the results of mixed model analysis difficult to interpret.

Response 3.4: The first 24 months of life (0-24 months of age), age-appropriate feeding practices and children’s nutritional status were measured at the same time in each follow-up. Then children’s nutritional status was measured for another 18 months. Generalized linear mixed-effect models capable to determine an association between age-appropriate feeding practices and chid growth over a repeated measurements. We do acknowledge that we cannot exclude reverse causality, we therefore use the term “association” between the determinants and outcomes (child growth).  

Point 3.5: Please describe your sensitivity analysis here.

Response 3.5: Our sensitivity analysis was used to determine the association between adherence to age-appropriate feeding practices and other potential determinants with child growth.

Selection criteria for sensitivity analyses were similar as our main analyses. The difference between them is that we also included children of which data collection with regard to the feeding practices was not fully complete (one or more follow-ups were missing). Generalized linear mixed-effects models were done. Fixed effects include regions, age, gender, age-appropriate feeding practices, caretaker’s education level, wealth index, child’s nutritional status, household main drinking water source, treatment of drinking water and ownership of latrine. Child’s ID with random slope of child’s age and nested by regions were selected as random effects. Age square was added into the models as age and child growth has nonlinear relationship (Appendix 2). Variables with p-value <0.1 was selected in the full model. The fitted models were compared by using Akaike information criterion (AIC) or Bayesian information criterion (BIC) estimates of model performance. An identity correlation matrix was chosen for the correlation among repeated child’s nutritional status measurements.

We found that child’s sex, child’s age, wealth index, caretaker’s education, hygiene practices, higher adherence to MMD and APF were associated with HAZ (shown in supplementary Table S1). Child’s age, sex, wealth index caretaker’s education, higher adherence to MDD and MAD were associated with WHZ (shown in supplementary Table S2); while only child’s age, wealth index and caretaker’s education were associated with WaSt (shown in supplementary Table S3).

Point 4: Results

Point 4.1: Is there any interaction between age and feeding practices as Figure 3 suggests that feeding practices change with age.

Response 4.1: There is an interaction between age and each age-appropriate feeding practices indicators in the bivariate analysis. However, in our generalized linear mixed-effect models, we did not add an interaction term between age and age-appropriate feeding practices into the model. The reason why we did not add the interaction term into the model was because age-appropriate feeding practices indicators were measured as a total sum score from the first 4 follow-ups. Hence, adding interact term between age and feeding practices indicators would not tell us that feeding practices change with age.

Point 4.2: This is a cohort study but occurrences of stunting, wasting and WaSt are interpreted as prevalence. I think it should be interpreted as incidence and prevalence is used for cross-sectional studies.

Response 4.2: We changed our interpreted “prevalence” to “percentage” through our results section. For example in line 279: “The prevalence of stunting….” Is now changed to “The percent of stunting..”.

Point 5: Table 2

Point 5.1: Please provide the number of observations used in each models. I guess they are different.

Response 5.1: We have inserted the number of included observations as the reviewer suggested. Table 2 is now changed to Table 4.

Point 5.2: Age is treated as a linear effect but Figure 2 suggests not.

Response 5.2: Age square is now added into the generalized mixed-effect models to deal with nonlinear between age and child growth.

Point 5.3: If P=0.000, please report as P<0.001

Response 5.3: All P=0.000 now is reported as P<0.001 as suggested.

Point 6: Table 3. See my comments on Table 2 and they apply here too.

 Response 6: We have inserted the number of included observations (observations=5,871) in Table 5 (previous version was Table 3) as the reviewer suggested.

Point 7: Table 4. See my comments on Table 2 and they apply here too.

Response 7: We have inserted the number of included observation (observations=5,871) in Table 6 (previous version was Table 4) as reviewer suggested.

Point 8: Discussion

Point 8.1: Please move the results of sensitivity analysis into Results section.

Response 8.1: The sensitivity analysis is now moved to the results section in line344-346, line 371-372 and line 390-392.

Point 8.2: Please mention the unobserved confounding issue and possible reversed causality issue in the limitations.

Response 8.2: We thank the reviewer for his/her comments. Environmental enteric dysfunction (EED) was suggested as one of the potential determinants for child growth by recent studies. Hence, a lack of EED information was then also added as limitation for this study in line 489-490. Child growth causes are complex and some of those factors could be potential confounders were beyond the scope of our study and it was added in line 490-492. Please also see our answer to your earlier comments on the reversed causality.

Reviewer 2 Report

Introduction (minor)

  1. First sentence of the introduction is quite long. I would consider splitting into two sentences
  2. “MMF is defined as the number of meals are consumed and reflects frequency.” Please remove the word “are”.

Methods

  1. First sentence please change to. “For out study, data was used from the parent study:”
  2. I would suggest adding a Figure which shows the timeline of follow-ups.
  3. Lines 97-98. Please change to: “For our study, only children aged 0-12 months at the 1st FU were selected to investigate their feeding practices for the first year of life.”
  4. Under child demography, I am getting a little confused because I thought the child was 12 months or younger at enrollment. So I would include some language that specifies what visit you are referring to.
  5. The hygiene practices section was very detailed, and I am not sure its importance for the manuscript at this point. Maybe include one sentence into why this is important for your paper.
  6. Feeding practices section. Please go into more detail into the 24 hour recall questionnaire. Does this only include how the child was feed in the last 24 hours. If so, you have a large limitation for exclusive breastfeeding since you are assuming they only received breastmilk for the time in between the FU visits. Also, please include if the child could receive things like vitamins, minerals, ORS for your exclusive breastfeeding definition
  7. I suggest putting the ADF criteria into a Table.
  8. Please remove the word period from this sentence. “Anthropometric measurements were collected for 30 months period from the 1st FU to the 6th FU.”
  9. Please reword the following sentence, “Weight, either length or height was measured twice per child to ensure the accuracy of the measurements.” To All anthropometric measurements, including weight, length, and height, were measured twice per child to ensure measurement accuracy.”
  10. Statistical analysis – second paragraph. First sentence is very long. Please consider making it into two sentences.

Tables and Figures

  1. The tables are of poor quality. Please consider creating them in word. Right now it looks like a blurry image.
  2. Figure 3 and 4. Please include a footer that defines what MDD, MMF, MAD, and ADF are. The figure should be able to stand alone, without having me look in the text to find what the acronyms mean.
  3. Figure 5. Would be great if you had each of the categories (i.e. stunting, wasting, and WaSt) add up to 100%. This will make it easier to compare to the other groups what the differences are between frequencies of each. Since wasting and WaSt have smaller total n’s, their percentages will obviously be smaller in each of the categories. My suggesetion will overcome this problem.
  4. Table 2. Only use 2 decimal places. Also, can’t have a p-value of 0. Please change these to <0.001
  5. Can you add figures from Tables 2, 3, and 4? It would be great to see the results depicted graphically.
  6. Please add to the appendix, a table showing characteristics between those who were lost to follow-up vs those who completed all FU. This will allow us to see if there is potential for selection bias.

Discussion

  1. Please include how the quality and quantity of food/diet can impact the gut microbiome. The gut microbiome is important for HAZ and WHZ, and this should be discussed.
  2. Please try not to draw too large of conclusions from your study. For example, “Moreover, interventions that aim to combat poor growth among these age groups should not be restricted to 24 months of age.” This is a strong statement for a non-intervention study. There are also a lot of confounders that were not included, so I would be careful with such powerful statements.
  3. We need more information on your sensitivity analysis. Did you do MI for those with missing information. A lot more information is needed on this before readers can feel comfortable with your statement about similar findings.
  4. I would also add the lack of information on child gut microbiota as a limitation.
  5. Please include information about breastfeeding in your conclusion.

Author Response

We would like to thank the reviewer for their time on reviewing our manuscript and providing useful comments and suggestions. Please find our responses point-by-point as following:

Response to Reviewer #2

Introduction:

Point 1: First sentence of the introduction is quite long. I would consider splitting into two sentences

Response 1: The sentence was split into two sentences as suggested in line 58. 

Point 2: “MMF is defined as the number of meals are consumed and reflects frequency.” Please remove the word “are”.

Response 2: The word “are” in line 77 was removed from the sentence as the reviewer suggested.

Methods:

Point 1: First sentence please change to. “For out study, data was used from the parent study:”

Response 1: The sentence was changed as reviewer suggested in line 100.

Point 2: I would suggest adding a Figure which shows the timeline of follow-ups.

Response 2: The timeline of follow-ups is now added in to the method section as Table 1.

Point 3: Lines 97-98. Please change to: “For our study, only children aged 0-12 months at the 1st FU were selected to investigate their feeding practices for the first year of life.”

Response 3.2: The sentence was changed as the reviewer suggested now in Line 116-117.

Point 4: Under child demography, I am getting a little confused because I thought the child was 12 months or younger at enrollment. So I would include some language that specifies what visit you are referring to.

Response 4: We thank the reviewer for his/her suggestion and we added “Child’s age was collected for 30 months from the 1st FU to the 6th FU.” in line 129 to specify that age of the child were collection for every data collection (1st -6th FU).

Point 5: The hygiene practices section was very detailed, and I am not sure its importance for the manuscript at this point. Maybe include one sentence into why this is important for your paper.

Response 5: We understand the reviewer’s concern and suggestion. Hygiene practices indicators is one of the important potential determinants for child growth, hence we think that it would be beneficial to reader to understand how these indicators were created. These methods were used by the National Institute of Statistics of the Ministry of Planning in Cambodia to report annually.  This information was also added in line 156-158.

Point 6: Feeding practices section. Please go into more detail into the 24 hour recall questionnaire. Does this only include how the child was feed in the last 24 hours. If so, you have a large limitation for exclusive breastfeeding since you are assuming they only received breastmilk for the time in between the FU visits. Also, please include if the child could receive things like vitamins, minerals, ORS for your exclusive breastfeeding definition

Response 6: Thank you for your suggestion. More detail on how the exclusive breastfeeding indicator was constructed was added in line 162-166. Caretakers were asked at what age their children started to receive liquid (other than breast milk) or other semi-liquid foods.  Caretakers of these children who received exclusive breastfeeding were asked if these children received any vitamins, minerals or oral rehydration salts. For breastfeeding practices, caretakers were asked in the 24hour recall. And therefore, exclusive breastfeeding would not be a confounder. Exclusive breastfeeding includes vitamins, minerals or oral rehydration salts and it was added in line 164.

Point 7: I suggest putting the ADF criteria into a Table.

Response 7: The table for construction of the ADF indicator was created and added in the manuscript as reviewer suggested. This table is now numbered as Table 2.

Point 8: Please remove the word period from this sentence. “Anthropometric measurements were collected for 30 months period from the 1st FU to the 6th FU.”

Response 8: The word “period” in line 205 was removed from the sentence as reviewer suggested.

Point 9: Please reword the following sentence, “Weight, either length or height was measured twice per child to ensure the accuracy of the measurements.” To All anthropometric measurements, including weight, length, and height, were measured twice per child to ensure measurement accuracy.”

Response 9: The sentence was changed according as the reviewer suggested in line 207-209.

Point 10: Statistical analysis – second paragraph. First sentence is very long. Please consider making it into two sentences.

Response 10: The sentence was split into two sentences as suggested in line 228.

Table and Figures:

Point 1: The tables are of poor quality. Please consider creating them in word. Right now it looks like a blurry image.

Response 1: All tables are now created in word as suggested.

Point 2: Figure 3 and 4. Please include a footer that defines what MDD, MMF, MAD, and ADF are. The figure should be able to stand alone, without having me look in the text to find what the acronyms mean.

Response 2: A footer that defines MDD, MMF, MAD, and ADF is now added into Figure 3 and Figure 4 as suggested by the reviewer. 

Point 3: Figure 5. Would be great if you had each of the categories (i.e. stunting, wasting, and WaSt) add up to 100%. This will make it easier to compare to the other groups what the differences are between frequencies of each. Since wasting and WaSt have smaller total n’s, their percentages will obviously be smaller in each of the categories. My suggestion will overcome this problem.

Response 3: Thank you for the suggestion. Figure 5 is now changed for each category of child growth (stunting, wasting and WaSt) and now adds up to 100% for the adherence to age-appropriate feeding practices indicator.

Point 4: Table 2. Only use 2 decimal places. Also, can’t have a p-value of 0. Please change these to <0.001.

Response 4: All P=0.000 is now changed to <0.001 throughout the paper as suggested.

Point 5: Can you add figures from Tables 2, 3, and 4? It would be great to see the results depicted graphically.

Response 5: We are also thankful for your suggestion on adding figures from Table 2, Table3 and Table 4 (please note that the number of the tables are now changed to Table 4, Table 5 and Table 6, respectively). Appendix 4 and 5 below show the prediction of HAZ and WHZ with age-appropriate feeding practices. Since concurrence wasting and stunting (WaSt) shows no association with any of age-appropriate feeding practices, these figures are not shown. For our study, we have include many tables, figures. Hence, the decision of whether to should add the two figures will be discussed with the editor of the Journal. 

Appendix 4: Prediction of child Height-for-Age Z score with total number of adherence to age-appropriate feeding practices among children with complete feeding practices.

MDD= minimum dietary diversity; MMF= minimum meal frequency; and ADF= age-appropriate daily feeding. Estimation fully adjusted linear mixed-effect models of HAZ on each age-appropriate feeding practices (MDD, MMF and APF) among children with complete information on feeding practices.

Appendix 5: Prediction of child Weight-for-Height Z score with total number of adherence to age-appropriate feeding practices among children with complete feeding practices.

MDD= minimum dietary diversity; and MAD=minimal acceptable diet; Estimation fully adjusted linear mixed-effect models of WHZ on each age-appropriate feeding practices (MDD and MAD) among children with complete information on feeding practices

Point 6: Please add to the appendix, a table showing characteristics between those who were lost to follow-up vs those who completed all FU. This will allow us to see if there is potential for selection bias.

Response 6: We created a table showing characteristics of child who were with or without complete information on feeding practices in the Appendix1. We also create two figures to show the characteristic among those children who were with or without complete information on feeding practices: percentage of child growth by age-groups among children who were with or without complete information on feeding practices (Appendix 2), and percentage of age-appropriate feeding practices by age among children with or without compete information on feeding practices (Appendix 3).

Discussion:

Point 1: Please include how the quality and quantity of food/diet can impact the gut microbiome. The gut microbiome is important for HAZ and WHZ, and this should be discussed.

Response 1: We thank the reviewer for his/her remark on the association between the gut microbiome and child growth. We now added this into our discussion section in line 441-447.

Point 2: Please try not to draw too large of conclusions from your study. For example, “Moreover, interventions that aim to combat poor growth among these age groups should not be restricted to 24 months of age.” This is a strong statement for a non-intervention study. There are also a lot of confounders that were not included, so I would be careful with such powerful statements.

Response 2: We would like to thank the reviewer for his/her kind comments and understand the reviewer’s concern. We now revised the statement in line 460-463.

Point 3: We need more information on your sensitivity analysis. Did you do MI for those with missing information. A lot more information is needed on this before readers can feel comfortable with your statement about similar findings.

Response 3: The criteria of our main analyses and sensitivity analyses is similar except the age-appropriate feeding indicators. For the main analyses we excluded children for who we did not have all data for feeding practices in the first 4 FUs. However, in the sensitivity analyses, we did include these children.  To be able to capture adherence of child feeding practices both quality and quantity of their food intake and breastfeeding, it would be crucial to have complete data on the feeding practices in their first 2 years of life. Sensitivity analyses were performed to prevent selection bias among our study children. 

For our sensitivity analysis we used generalized linear mixed effect models to determine the association between adherence to age-appropriate feeding indicators and other potential determinants with child growth.

We now added more details on the methods of sensitive analyses in method section line 24-256.We did not do the MI for those with missing information. Since the adherence of each age-appropriate feeding practices were created by sum of each scores per each age-appropriate feeding practice indicators, it would lead to a complication in interpretation the results. Also, as show in the Appendix 1, 2 and 3, children who were not lost in any FU and children who were lost in some FUs share very similar characteristics to each other.

Point 4: I would also add the lack of information on child gut microbiota as a limitation.

Response 4: We added the lack of information on EED as one of our limitation in line 425- 246.

Point 5: Please include information about breastfeeding in your conclusion.

Response 5: The information on breastfeeding is now including in our conclusion as suggested in line 497 and line 501.

Round 2

Reviewer 1 Report

1. Abstract and Figure 1

"Children (n=1079) were followed for 30 months (6 follow-ups)." but only 1018 children were included in the analysis. 1079 children were included in the baseline data analysis. Please make this point clear in the Abstract and Figure 1. 

2. Abstract

"0.25and" should be "0.25 and"

3. Figure 3

"age-appropraite feeding pratices" should be "age-appropriate feeding practices"
